# Coxsackievirus A6 Infection Causes Neurogenic Pathogenesis in a Neonatal Murine Model

**DOI:** 10.3390/v15020511

**Published:** 2023-02-12

**Authors:** Qiang Sun, Jichen Li, Rui Wang, Tiantian Sun, Yanjun Zong, Congcong Wang, Ying Liu, Xiaoliang Li, Yang Song, Yong Zhang

**Affiliations:** 1WHO WPRO Regional Polio Reference Laboratory, National Health Commission Key Laboratory for Biosafety, National Health Commission Key Laboratory for Medical Virology, National Institute for Viral Disease Control and Prevention, Chinese Center for Disease Control and Prevention, 155 Changbai Road, Beijing 102206, China; 2Center for Biosafety Mega-Science, Chinese Academy of Sciences, Wuhan 430071, China

**Keywords:** coxsackievirus A6, animal model, RNA-Seq, pathogenesis

## Abstract

Coxsackievirus A6 (CVA6), a member of species A enterovirus, is associated with outbreaks of hand-foot-and-mouth disease and causes a large nationwide burden of disease. However, the molecular pathogenesis of CVA6 remains unclear. In the present study, we established a suckling Institute of Cancer Research (ICR) mouse infection model to explore the neural pathogenicity of CVA6. Five-day-old mice infected with CVA6 strain F219 showed lethargy and paralysis, and died 5 or 6 days after infection via IM injection. Cerebral edema and neuronal cell swelling were observed in the infected brain tissue, and we found that the CVA6 VP1 antigen could co-localize with GFAP-positive astrocytes in infected mouse brain using an immunofluorescence assay. CVA6 strain F219 can also infect human glioma (U251) cells. Transcriptome analysis of brain tissues from infected mice and infected U251 cells showed that significantly differentially expressed genes were enriched in antiviral and immune response and neurological system processes. These results indicate that CVA6 could cause neural pathogenesis and provide basic data for exploring the mechanism of how host–cell interactions affect viral replication and pathogenesis. Importance: Coxsackievirus A6 (CVA6) surpasses the two main pathogens, enterovirus 71 (EV-A71) and coxsackievirus A16 (CVA16), which are the leading pathogens causing HFMD in many provinces of China. In our study, CVA6 infection caused neurogenic pathogenesis in a neonatal murine model, manifesting as cerebral edema and neuronal cell swelling, CVA6 VP1 antigen could co-localize with GFAP-positive astrocytes in the infected mouse brain. Based on CVA6-infected brain tissue and U251 cell transcriptome analysis, we found upregulated antiviral and immune response-related genes such as Zbp1, Usp18, Oas2, Irf7, Ddx60, Ifit3, Ddx58, and Isg15, while the neurological system process-related genes were downregulated, including Fcrls, Ebnrb, Cdk1, and Anxa5.

## 1. Introduction

Enteroviruses (EVs) are found commonly worldwide and cause various diseases such as hand, foot, and mouth disease (HFMD), viral encephalitis, aseptic meningitis, and acute flaccid paralysis [1]. It is worth noting that out of the annual reported HFMD cases in mainland China since 2013, the proportion caused by coxsackievirus A6 (CVA6) has increased every year, and at present, has surpassed enterovirus 71 (EV-A71) and coxsackievirus A16 (CVA16), the leading pathogens in many provinces of China [2,3]. CVA6-related HFMD presents several clinical features, such as atypical herpes and nail removal, which are sometimes complicated by severe clinical symptoms such as meningitis and encephalitis [4,5,6,7]. However, no vaccine or effective drugs are present [8], and the pathogenesis of CVA6-associated diseases such as HFMD, meningitis, and encephalitis is unclear [9]. 

Several previous studies have reported that CVA6 murine models using one- to ten-day-old BALB/c or Institute of Cancer Research (ICR) mouse strains through IP, IM, and IC injection exhibited neurotropism and triggered systemic manifestations [8,9,10,11]. However, changes in biological processes at the tissue level and in their regulation at the molecular level during CVA6 infection are rare. Several studies have analyzed the transcriptomic expression profiles of EV infections, mainly EV-A71 and CVA16, which provide clues regarding the pathogenic mechanisms.

In the present study, we established a neonatal mouse model of CVA6 infection for CVA6-related neurological disease research. In addition, we used this model to describe the neurogenic pathogenesis of CVA6 and describe the dysregulation at the molecular level in mouse brain tissue and human glioma (U251) cell level via RNA sequencing (RNA-Seq). Our study established a 5-day-old CVA6-infected ICR mouse model, and provided basic data for exploring the mechanism of host–cell interaction affecting viral neurogenic pathogenesis.

## 2. Results

### 2.1. Establishment of a Neonatal ICR Mouse Model of CVA6 Infection

In this study, to establish a mouse model for studying CVA6 pathogenesis, we selected 5-day-old ICR mice through IC, IM, or IP injection, to inoculate CVA6 with serially diluted 10-fold (10^3^ to 10^6^ 50% tissue culture infective dose [TCID_50_] per mouse) (*n* = 7 per group) and one negative group (*n* = 7 per group). Their clinical symptoms were observed, and the differences among the three routes of infection were compared. 

The average body weights of the mice in the experimental groups infected with 10^5^–10^6^ TCID_50_ of CVA6 were significantly different from those of the mice in the 10^3^–10^4^ TCID_50_ of CVA6 and the negative control groups via all three injection routes (Figure 1A–C). All mice infected with 10^5^ to 10^6^ TCID_50_ via the IM route died 5 to 6 days post infection (dpi); however, under these two doses, some infected mice died after IC and IP routes. At the 10^4^ TCID_50_ infection dose level, some infected mice died after IM and IC routes, and the mortality of the former was higher than that of the latter; nevertheless, all infected mice survived via the IP route. At the 10^3^ TCID_50_ infection dose level, comparing all the infected mice that survived via IC and IP route, only small partially infected mice died via IM route (Figure 1D–F). Moreover, the average clinical scores tended to be consistent with average body weight and survival rates. All mice infected with 10^5^–10^6^ TCID_50_ via the IM route developed significant hindlimb paralysis, gradually lost weight, and died within 5–6 dpi, with a clinical score of 5 between 5–6 dpi (Figure 1G–I). These results indicated that mice infected with 10^5^ to 10^6^ TCID_50_ via the IM route were suitable for the infection mouse model with a stable time of disease onset, mortality rate, and good reproducibility. Therefore, in this study, we selected 10^5^ TCID_50_ via the IM route for further investigation.

However, under these two doses, some infected mice infected by IC and IP routes died.

### 2.2. Effective Replication and Pathological Changes of CVA6 in Infected Mice Brain and Muscular Tissues

The viral loads in 5-day-old mice inoculated with 10^5^ TCID_50_ via the IM route were measured at 1, 3, and 5 dpi. The results showed that the viral loads in the muscular tissues ranged from 10^3.5^ TCID_50_ at 1 dpi to 10^6.25^ TCID_50_ at 5 dpi, which was consistent with many other studies indicating the muscular tissues, the strongest tissue tropism of EVs (Figure 2A). However, the viral loads in the brain tissues ranged from 10^1.25^ TCID_50_ at 1 dpi to 10^3.75^ TCID_50_ at 5 dpi, which indicated effective replication of CVA6 in suckling mouse brain (Figure 2B).

Subsequently, these two infected mouse tissues at 5 dpi underwent hematoxylin and eosin (H&E) staining for analysis of histopathological changes. Compared with the control group, in infected mice, hind limb skeletal muscle fibers exhibited severe necrosis and rupture, as well as inflammatory edema (Figure 2C–E). The encephalitis symptoms present in infected mice with cerebral edema and neuronal cell swelling were similar to the neurological symptoms in humans (Figure 2F–H).

### 2.3. CVA6 Antigen Could Co-Localize with GFAP-Positive Astrocytes in Infected Mice Brain

The pathological manifestations of fatal cases caused by CVA6 are characterized by inflammatory damage to the central nervous system (CNS) [12]. Many studies have shown that astrocytes are emerging as pivotal regulators of CNS inflammatory responses [13,14]. GFAP (glial fibrillary acidic protein) is a class-III intermediate filament, a cell-specific marker, which differentiates astrocytes from other glial cells during the development of the central nervous system. Thus, in this study, we conducted immunofluorescence analysis of the potential targeted cells in the infected mouse brain [15]. Immunofluorescence staining studies showed that compared to NeuN-positive neuronal cells, the CVA6 VP1 antigen co-localized with GFAP-positive astrocytes (Figure 3A,B). This result indicates that the potential preferential infection of astrocytes by CVA6 may play a key role in viral neurological pathogenesis. Therefore, the exact molecular mechanisms should be further explored.

### 2.4. CVA6 Strain F219 Replicates Well in Human Glioma (U251) Cells

Since we found that the CVA6 VP1 antigen could co-localize with GFAP-positive astrocytes in the infected mouse brain, we further determined whether CVA6 could replicate in human neuronal cell lines. We measured viral replication dynamics in the human glioma cell line U251 and measured the viral titer at five time points after infection with a multiplicity of infection (MOI) of 5. CVA6 could cause cytopathic effects (CPE) after 24 h of infection and was replicated well in U251 cells via growth curves (Figure 4A,B). Consistent with the growth curve, immunofluorescence staining revealed that the CVA6 VP2 capsid protein could be detected in U251 cells and that it co-localized with GFAP (Figure 4C).

### 2.5. CVA6 Infection Increases the Defense Response to Virus Genes and Decreases the Number of Neurological System Genes in the Brain and U251 Cells at the Transcriptome Level

The above results demonstrated the dynamic virus titers and pathological changes at both the infected mouse brain tissue and human glioma (U251) cell levels. However, the molecular mechanisms underlying CVA6 pathogenesis are not yet fully understood. It is well known that viruses act as strictly parasitic organisms, and investigation of the mechanism of how host–cell interactions affect viral replication and pathogenesis is essential.

In this study, we described CVA6 infection, especially changes in differential host factor expression after infection, in human glioma (U251) cells and infected mouse brain using transcriptome sequencing analysis (Figure 5A). We used Hiplot Enhanced MA v0.1.0 analysis, which showed that the significantly differentially expressed genes (DEGs) were interferon-stimulated genes (ISGs), such as Ifit1, GBP4, etc., and inflammatory factors, such as CXCL10, Stat1, etc., which indicated that CVA6 maintained effective infections in human glioma (U251) cells and mouse brain (Figure 5B,C). The Gene Ontology (GO) and Kyoto Encyclopedia of Genes and Genomes (KEGG) pathway enrichment analyses of DEGs in the RNA-Seq of infected U251 cells showed that antiviral or inflammation-related signaling pathways, such as Type I interferon signaling pathway, toll-like receptor signaling pathway, cytokine–cytokine receptor interaction, and so on, and the neurological system, process pathway enrichments (Figure 5D,E). RNA-Seq of infected mouse brain showed antiviral or inflammation-related signaling pathways, such as the Type I interferon signaling pathway, RIG-I-like or Toll-like receptor signaling pathway, NF-kappa B signaling, and other neurological complications, such as neuroactive ligand–receptor interaction, neuron projection, glioma, etc. (Figure 5F,G). Notably through further visualization analysis of mutual DEGs, we found that some genes associated with defense response to viruses pathway (Oas2, Dtx3l, Gbp7, Ddx58, Usp18, Rtp4, Ifit3, Nlrc5, Rsad2, Oasl1, Irf7, Bst2, Trim34a, Ddx60, Isg15, Ifi204, Slfn8, Mx1, Stat1, Ifitm3, Parp9, Ifit1, Zbp1, and Itgam) were significantly upregulated, while most genes associated with neurological system process pathway (Rasgrp3, Abca9, Gpr37l1, Pla2g7, Fut9, Aqp4, Ednrb, Cdk1, Cenpf, Top2a, Hmgb2, S100b, Anxa5, Hebp5, and Sptssb) were significantly downregulated compared to the other three genes associated with neuro-inflammation pathways (Mertk, C1qb, and Fcrls) which were significantly upregulated (Figure 5H).

Taken together, we conclude that neurotropic CVA6 preferentially infects and replicates in astrocytes of the neonatal murine model. CVA6 could cause pathogenesis in the ICR mouse brain tissues and the human glioma (U251) cells to the detailed dysregulated genes with notable features showing upregulation of antiviral and immune response-related genes such as Zbp1, Usp18, Oas2, Irf7, Ddx60, Ifit3, Ddx58, and Isg15, while downregulating neurological system process-related genes, such as Fcrls, Ebnrb, Cdk1, and Anxa5.

## 3. Discussion

In recent years CVA6 has caused HFMD epidemics in mainland China have become a serious public health concern with a large number of severe cases [12]. Nevertheless, there is no effective vaccine or drug for CVA6 infection. Although many studies have reported the clinical medicine associated with central nervous system disease and epidemiological studies of CVA6 characterized by highly frequent recombination events correlated with pathogenicity, the molecular pathogenesis is deemed insufficient [2,6,13,14]. Animal models can be used to explore the pathogenic mechanisms of viral infection; since suckling murine and primate animals are suitable hosts for human enteroviruses, previous studies have shown that animal models can be used to explore possible mechanisms of enterovirus infection [15,16,17,18]. Several previous studies have reported CVA6 murine models of one- to ten-day-old BALB/c or ICR mouse strains. In this study, we established a 5-day-old CVA6-infected ICR mouse model and described the neurogenic pathogenesis. We conclude that neurotropic CVA6 preferentially infects and replicates in astrocytes of the neonatal mouse model, and also replicates well in human glioma (U251) cells. Finally, CVA6 infection increases the defense response to viral genes and reduced the number of neurological system genes in the brain and U251 cells at the transcriptome level.

CVA6 infection can cause neurological and systemic complications such as HFMD, aseptic meningitis, and encephalitis [9,10,11]. In this study, the brain edema, neuronal cell swelling, and high viral titers in the infected mice also led to significant astrocyte proliferation, microgliosis, and immune cell aggregation [19]. Previous studies have also shown that enterovirus 71 (EV-A71) preferentially infects astrocytes, and that echovirus 11 or 30 can infect human glioma (U251) cells, but not the neuroblastoma cell line SK-N-SH [20,21,22]. We also found CVA6 antigen co-localization with GFAP-positive astrocytes, which may be consistent with the enterovirus-infected brain leading to gliosis [23]. Further studies are needed to explore the consequences of neuronal death and dysfunction.

Previous studies have reported that EV-A71, CVA16, and echovirus 30 can result in the differential expression of genes in the nervous system development pathways [21,24,25,26]. In this study, CVA6 infected 5-day-old ICR mice by intramuscular injection and human glioma (U251) cells, and differentially expressed genes (DEGs) were screened using RNA-seq analysis. The results of this study indicate that after CVA6 infection, DEGs that are enriched in neurological system processes, such as Fcrls, Ebnrb, Cdk1, and Anxa5, which play important roles in glial cell growth and migration, were significantly downregulated. Astrocytes secrete key proteins and metabolites used to fuel neurons as well as in the biogenesis of neurotransmitters [27,28]. EV infection systematically causes neural inflammation and injury in the brain via dysfunctional astrocytes that disrupt brain homeostasis [29].

However, DEGs were enriched in antiviral and immune responses that were upregulated in response to CVA6 infection. Our analysis showed that Zbp1, Usp18, Oas2, Irf7, Ddx60, Ifit3, Ddx58, and Isg15 were significantly upregulated after CVA6 infection. CVA6 infection leads to the increased expression of ISGs, including Oas2, Irf7, Ddx60, Ifit3, Ddx58, and Isg15, which exhibit immunopathogenic potential and are implicated in neural inflammation [30]. Z-DNA-binding protein 1 (Zbp1) can be induced by type I interferons (IFNs) to upregulate IFN expression after viral infection, thus activating the expression of interferon regulatory factor 1 (IRF1) [31]. In addition, Zbp1 was activated by the receptor-interacting serine/threonine-protein kinase (RIPK) signaling pathway to alter neuronal cell metabolism and inhibit viral genome replication [32]. Previous studies have shown that ubiquitin specific peptidase 18 (Usp18) is involved in the negative regulation of IFN-induced inflammation and is enriched in the mitochondria of virus-infected cells to promote the polyubiquitination and aggregation of mitochondrial antiviral signaling protein (MAVS) [33]. This also provides insight into the specific mechanisms of CVA6 infection. Since the innate immune response is the first line of viral defense, our transcriptome sequencing data revealed dysregulation of neuropathy-related pathways [34]. Therefore, we used a five-day-old mouse model to clarify the pathological characteristics of CVA6 and found that antiviral and immune responses and neurological system process pathways could be involved in the pathogenesis of CVA6 at the molecular level. These findings provide a theoretical basis for understanding the pathogenic mechanisms.

## 4. Materials and Methods

### 4.1. Ethics Statement

ICR mice were purchased from SPF Biotechnology Co., Ltd. (Beijing, China). All animal experiments were approved by the Ethics Review Committee of the National Institute for Viral Disease Prevention and Control, Chinese Center for Disease Control and Prevention (approval No. 20201022059).

### 4.2. Virus, Cells, and Mice

One CVA6 clinical isolate, F219, was isolated from a severe case during an outbreak of HFMD in Yunnan in 2016. In addition to the typical clinical manifestations of HFMD, such as fever, rash on hands and feet, this patient also presented some nervous system symptoms, such as headache, poor spirit, lethargy, etc.

Human rhabdomyosarcoma (RD; ATCC) cells were cultured in Dulbecco’s modified Eagle medium (DMEM) (Gibco, CA, USA) containing 10% fetal bovine serum (FBS) (Gibco, USA) and 1% penicillin-streptomycin (HyClone, Logan, UT, USA) at 37 °C with 5% CO_2_. In addition, human glioma cells (U251; ATCC) were cultured in a 1:1 mixture of DMEM and F-12 (Gibco, California, USA), supplemented with 10% FBS. After the degree of cell fusion reached 80–90%, the medium was replaced with medium containing 2% FBS (maintenance media; MM) and incubated with the virus to obtain CPE. After three freeze-thaw cycles, all viruses in the supernatant were filtered through a 0.2-mm filter and stored at −80 °C.

All ICR mice were raised in independently ventilated cages and were provided with sufficient food and water.

### 4.3. Animal Infection Experiments

Five-day-old ICR mice were infected with CVA6 strain F219 at 10^3^, 10^4^, 10^5^, or 10^6^ 50% TCID_50_ through IC, IM, and IP routes (*n* = 7 per group); control mice were inoculated with MM and reared separately from the infected mice. The weights, survival rates, and clinical scores of the mice were recorded daily. The grade of clinical disease was scored as follows: 0, no disease; 1, ruffled fur; 2, weight loss; 3, single limb paralysis; 4, hind limb paralysis; and 5, moribund, or dead [35].

### 4.4. Virus Isolation and Titers in Mouse Tissues

Brain and hind limb muscles were harvested from 5-day-old ICR mice via IM injection of F219 at 1, 3, and 5 dpi (*n* = 5 per time point). All samples were added to phosphate-buffered saline containing 1% penicillin-streptomycin and crushed in a tissue grinder (Scientz, Ningbo, China) for three freeze-thaw cycles before centrifugation to obtain the supernatant. The supernatant collected at each time point was serially diluted 10-fold and inoculated into RD cells that were cultured in 96-well plates. After 5 days of continuous observation, virus titers were detected using the TCID_50_ assay.

### 4.5. Growth Curves

U251 cells were infected at an MOI of 5 with the CVA6 for multicycle replication under the conditions of 5% CO_2_ and 37 °C. After 2 h of incubation, the cells were washed three times with PBS and the infection medium was added. At 2, 6, 12, 24, and 48 h post infection, supernatants were collected, and viral titers were determined in RD cells.

### 4.6. Histopathological and Immunofluorescence Assay

After IM injection of F219 in 5-day-old ICR mice, the brains and hind limb muscles of the experimental and control groups were harvested at 5 dpi. The tissues were fixed in formalin buffer for 24 h, dehydrated, embedded in paraffin, and sliced into 5-μm-thick sections. H&E staining was performed after deparaffinization of the tissue sections with xylene. Finally, all images were observed using an AxioCam MRc5 (Carl Zeiss, Berlin, Germany) at magnifications of 200 or 400×.

For immunofluorescence analysis, tissues were fixed in 10% paraformaldehyde solution for 24 h, embedded in optimal cutting temperature compound (OCT), and sectioned. OCT-embedded sections were stained to evaluate CVA6 VP1 co-localization with NeuN-positive neurons or GFAP-positive neuroglial cells. Rabbit polyclonal anti-VP1 CVA6 antibody (1:100 dilution; GeneTex, San Antonio, TX, USA, Cat No. GTX132346, USA), mouse monoclonal anti-GFAP antibody (1:100 dilution; Abcam, Cambridge, UK, Cat No. ab4648, United Kingdom), and mouse monoclonal anti-NeuN antibody (1:100 dilution; Abcam, Cat No. ab104224, United Kingdom) were incubated at 4 °C overnight.

The CVA6-infected U251 cells were detected using rabbit polyclonal anti-VP1 of the CVA6 antibody (1:100 dilution; GeneTex, Cat No. GTX132346, USA) and mouse monoclonal anti-GFAP antibody (1:100 dilution; Abcam, Cat No. ab4648, United Kingdom). Fluorescence Dylight 680, Goat Anti-Rabbit IgG and Dylight 488, and Goat Anti-Mouse IgG antibodies (1:500 dilution) were added and incubated at room temperature (20 °C) for 1 h. Finally, all images were observed using a Leica laser scanning confocal microscope (Leica, Wetzlar, Germany) at magnifications of 200× or 400×.

### 4.7. Library Preparation for Transcriptome Sequencing

CVA6 was infected via IM inoculation with 10^5^ TCID_50_ in 5-day-old ICR mice and in the control group (*n* = 3 per group). TRIzol reagent was used to extract total RNA from the mouse brain tissues. Qubit 2.0 (Invitrogen, Waltham, MA, USA) was used to accurately quantify RNA concentrations and Bioanalyzer 2100 (Agilent, Santa Clara, CA, USA) was used to detect RNA integrity number (RIN ≥ 8). Briefly, the first strand of cDNA was synthesized using random primer reverse transcription and the second-strand cDNAs were amplified using RNase H and DNA polymerase. Finally, the enriched and purified cDNAs were sequenced on an Illumina NovaSeq 6000 platform. Gene expression levels were quantified using a feature count (version 1.5.0).

### 4.8. RNA-Seq Data Bioinformatics Analysis

The Deseq2 software was used for differential gene expression analysis, and a significance cut-off of *p* < 0.05 and Log2 fold change ≥ ±1.5 was used as the standard for screening differentially expressed genes. To clearly understand the biological significance of differentially expressed genes, GO and KEGG pathway analyses were performed. GO analysis was carried out for three aspects: cellular component (CC), biological process (BP), and molecular function (MF). KEGG pathway analyses were used to correlate the catalog of differentially expressed genes with system functions at higher levels of cells, species, and ecosystems to better understand the molecular response networks of protein-coding genes. These were then enriched and analyzed using the online tool Database for Annotation, Visualization, and Integrated Discovery (DAVID) Bioinformatics Resources 6.8.

### 4.9. Statistical Analysis

All statistical analyses were performed using GraphPad Prism 8.0 (GraphPad Software). The survival rates of different groups of mice were assessed using the log-rank (Mantel–Cox) test. The Wilcoxon test was used to analyze the clinical score curves. The tissue viral titers in different mice were analyzed by two-tailed Student’s *t*-tests, and *p* values < 0.05 were considered significant.

## Figures and Tables

**Figure 1 viruses-15-00511-f001:**
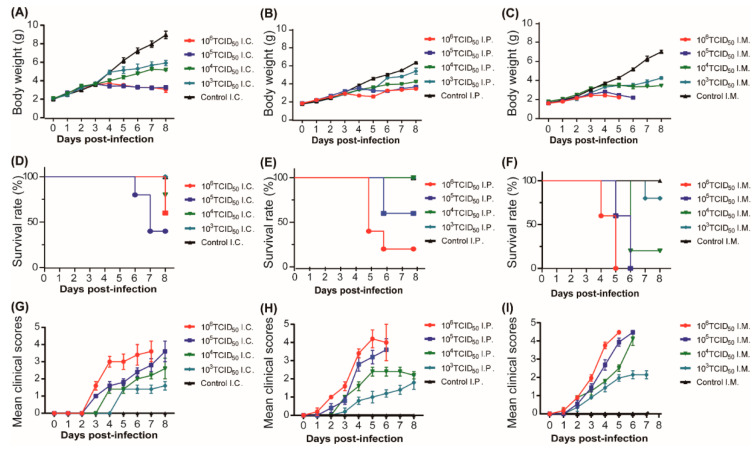
Body weight, survival rate, and clinical score of suckling mice in inoculation route and F-virus challenge dose experiments. Five-day-old ICR mice were infected with strain F219 at 10^3^, 10^4^, 10^5^, and 10^6^ TCID_50_ by intracranial injection (**A**,**D**,**G**), intraperitoneal injection (**B**,**E**,**H**), and intramuscular injection (**C**,**F**,**I**), respectively (*n* = 7 per group); control mice were inoculated with cell culture maintenance medium (MM). Body weight (**A**–**C**), survival rate (**D**–**F**) and clinical scores (**G**–**I**) of 5-day-old mice inoculated via IC, IP, and IM were monitored daily. Clinical scores of CVA6-infected and control mice. 0, no disease; 1, ruffled fur; 2, weight loss; 3, single limb paralysis; 4, hindlimb paralysis; 5, moribund or dead.

**Figure 2 viruses-15-00511-f002:**
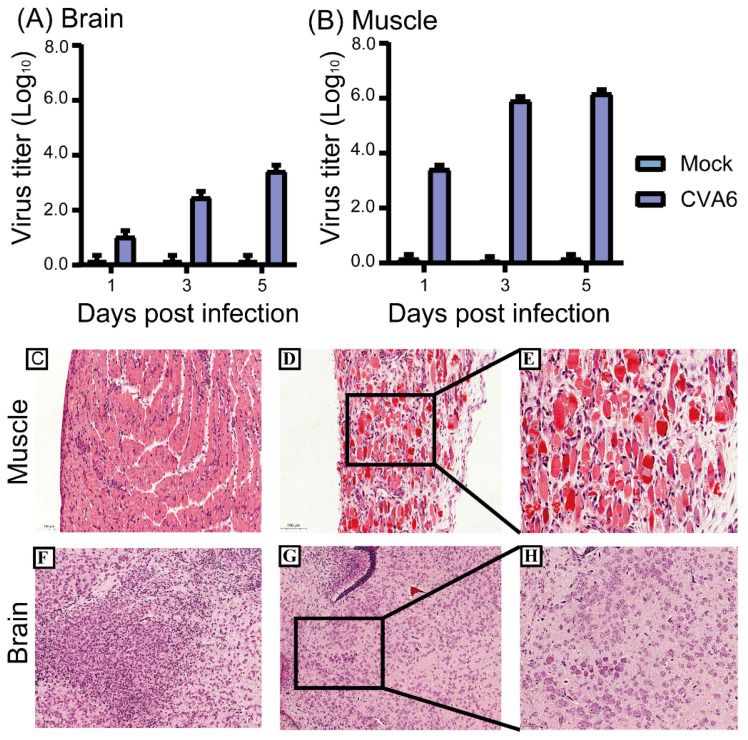
Virus titers in tissues and histopathological changes of 5-day-old mice. Intramuscular injection of the CVA6 strain F219 was performed in 5-day-old ICR mice. Mice were then sacrificed after 1, 3, and 5 days, and virus titers were measured in the brain and muscle tissue (**A**,**B**). Histopathological analysis showing muscle fiber rupture (**D**,**E**), cerebral edema, and neuronal cell swelling (**G**,**H**); no changes were observed in control tissues (**C**,**F**). The magnification of (**C**,**D**,**F**,**G**) is × 200, and the magnification of (**E**,**H**) is ×400. The scale bar for (**C**,**D**,**F**,**G**) is 100 μm, and the scale bar for (**E**,**H**) is 50 μm.

**Figure 3 viruses-15-00511-f003:**
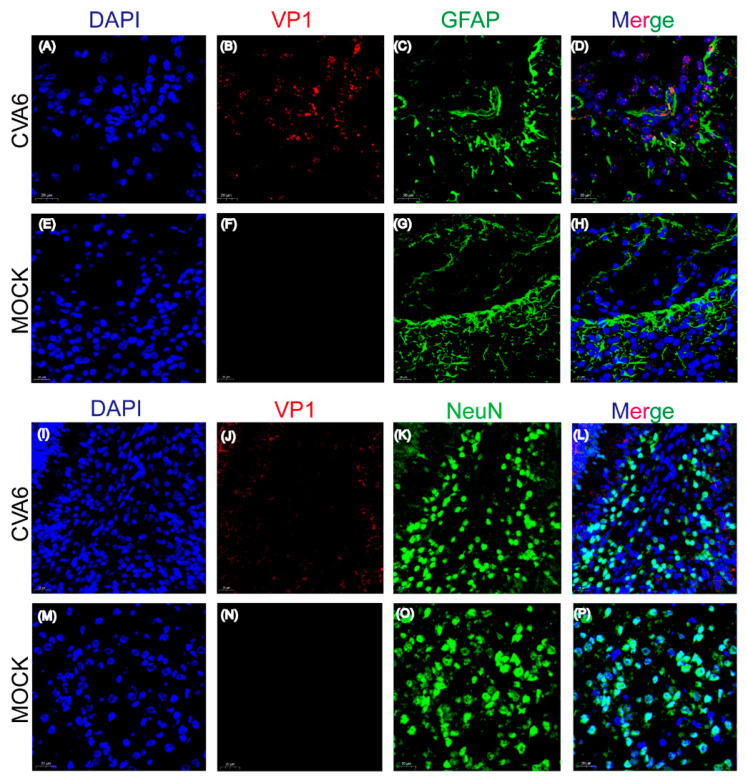
Co-localization of viral antigens with astrocyte marker GFAP in infected brain tissues. Suckling mice were injected with CVA6 or cell culture maintenance medium. Brain tissue sections of suckling mice were fixed and immunostained with DAPI (Blue), VP1 (Red), and astrocyte marker GFAP (Green) (**A**–**H**); immunostaining with DAPI (Blue), VP1 (Red), and neuron marker NeuN (Green) (**I**–**P**); Magnifications: (**A**,**B**) ×400. Scale bar = 20 μm.

**Figure 4 viruses-15-00511-f004:**
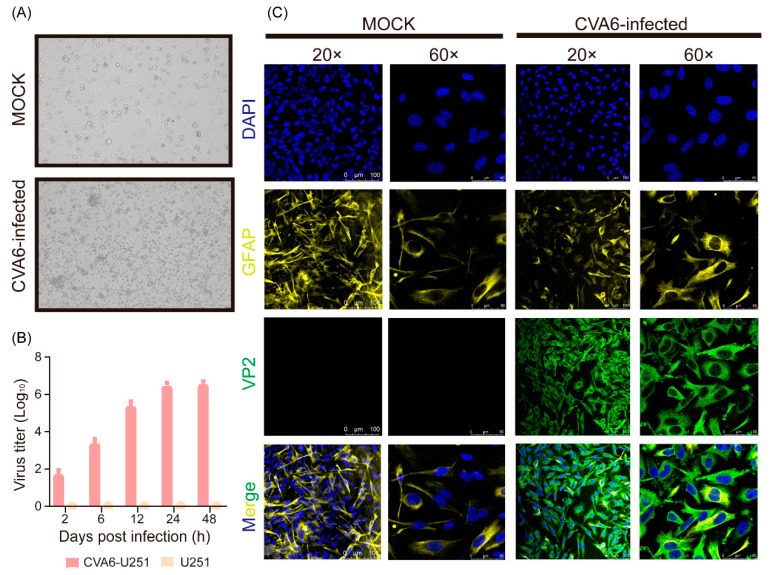
CVA6 infects U251 cells. Cytopathic effect of virus on U251 at 24 h post infection (**A**); cells were inoculated with CVA6 at 37 °C, and harvested at 2, 6, 12, 24, and 48 h post-inoculation. Virus titers were determined on RD cells (**B**); U251 cells infected with CVA6 and control were fixed and immunostained with DAPI (Blue), astrocyte marker (GFAP, Yellow), and VP2 (Green). The representative images were acquired using fluorescence confocal microscopy. Magnifications: (**A**,**B**), ×200 × 600. Scale bar = 100 μm (**C**).

**Figure 5 viruses-15-00511-f005:**
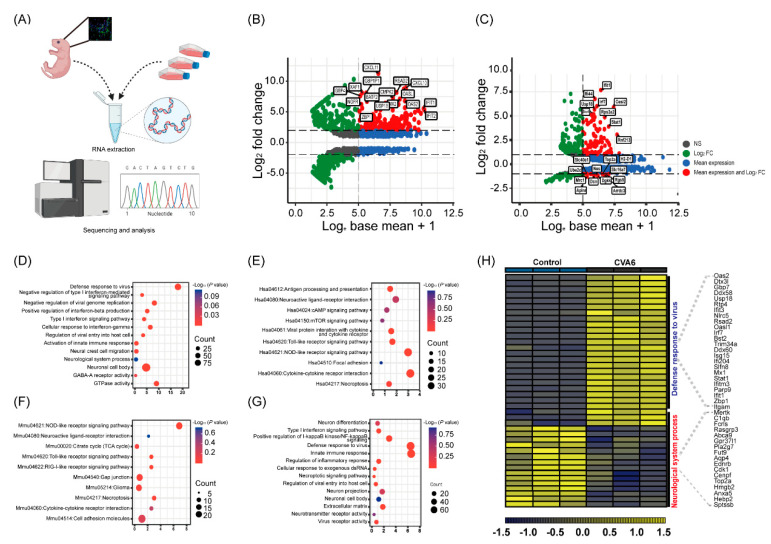
Transcriptome analysis of the brain tissues and U251 cells infected with CVA6. Total RNA was extracted from U251 cells and brain tissues of suckling mice and analyzed by sequencing (**A**). Hiplot Enhanced MA v0.1.0 analysis showing the comparison of the RNA-Seq of differential gene expression of infected U251 cells (**B**) and that of infected suckling mice brain tissue (**C**). GO and KEGG pathway enrichment analysis of the RNA-Seq of differentially expressed genes in the infected U251 cells (**D**,**E**) and that of infected brain tissue (**F**,**G**). The dot plot displays the enrichment score values. Visualization of differentially expressed genes in the brain tissue (**H**).

## Data Availability

All sequencing data in this study are available in the Sequence Read Archive (SRA) database under the accession numbers PRJNA929099 and PRJNA929115.

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
