# Peer review of "Coxsackievirus A6 Infection Causes Neurogenic Pathogenesis in a Neonatal Murine Model"

_viruses, 2023, doi:10.3390/v15020511_

Round 1

Reviewer 1 Report

Sun et al. developed a neonatal mouse model for studying the pathogenesis of CVA6 infection in neurons, with clear clinical and immunological phenotypes in infected mice. The experiments are well controlled, and the data is mostly clearly presented. The model developed here will be useful to people working with CVA6 and other similar neurotropic pathogens.

I only have a few minor comments:

- The manuscript in general would benefit from copy-editing or proof-reading to correct grammatical errors and make it more coherent.

- The sentence on line 65 to 68 is not clear

- In figure 1 (line 87), it not clear which of the graphs represent which route of inoculation (intracranial, intramuscular, intraperitoneal, etc). It is implied in the figure legend, but it needs to be clearly indicated in the figure.

- It is not clear what "partial infection", referred to in line 75, means.

- The use of "WT" to denote mock-infected mice in figures 2A and 2B is confusing, since "WT" is often used to mean "wild type".

- The legend for figure 2 says the scale bar is 100um, but that is only true for figure 2D. The scale bar for figure 2C appears to be 500um.

- References need to be provided for statements such as those in line 113-115, 117-118, 118-119, etc

- It is not clear what the "20x" refers to in figure 3 (line 126). Is that the objective lens used? If it is, I am not sure it needs to be indicated here.

- It is not clear whether the heat map in figure 5H represents data from the mice tissues or the U251 cells. The legend suggests that it is both, but I don't see the point of merging them, if that is the case. In any case, the data from the brain tissue needs to be highlighted, given that the main purpose of this manuscript is to describe the mouse model being established.

- The statement "CVA6 could cause pathogenesis at both levels to the detailed dysregulated genes..." in line 190-191 is not clear

- In the Discussion section, the authors wrote "Several previous studies have reported CVA6 murine models of one- to ten-day-old BALB/c or ICR mouse strains" (line 210-211). Given that at least one previous study has shown norological and immonological phenotypes in 10-day old ICR mice (PMID: 36036059), I suggest the authors follow their statement with an explanation of how their model is superior to the previously published ones, or perhaps how it is different, what advantages it has, etc.

- In line 211-215, the authors state "CVA6 infection leads to neurological and systemic complications such as HFMD, aseptic meningitis, and encephalitis (9-11). In this study, the brains of infected mice showed cerebral edema, neuronal cell swelling, and high viral titers. The virus, including EV-A71, infects the human brain and has been shown to cause marked astrogliosis, microgliosis, and immune cell accumulation". This makes it sound like EV-A71 is a subset of CVA6, which is not the case.

- In line 244-246, the authors state "...is enriched in virally infected mitochondria to promote the polyubiquitination and aggregation of mitochondrial antiviral signaling protein (MAVS)". I imagine the authors mean 'enriched in the mitochondria of virus-infected cells', since viruses don't typically infect mitochondria.

- RNA-seq data needs to be made publically available, such as by depositing it in a public repository.

Author Response

Reviewer 1

Sun et al. developed a neonatal mouse model for studying the pathogenesis of CVA6 infection in neurons, with clear clinical and immunological phenotypes in infected mice. The experiments are well controlled, and the data is mostly clearly presented. The model developed here will be useful to people working with CVA6 and other similar neurotropic pathogens.

I only have a few minor comments:

- The manuscript in general would benefit from copy-editing or proof-reading to correct grammatical errors and make it more coherent.

------ Response to reviewer 1 comment 1: The manuscript has been proofread by a native English speaker to correct grammatical errors and make it more coherent.

- The sentence on line 65 to 68 is not clear

------Response to reviewer 1 comment 2: Thank you for your comments, and the sentence was revised as “In addition, we used this model to describe the neurogenic pathogenesis of CVA6, and describe the dysregulation at the molecular level in mouse brain tissue and human glioma (U251) cell level via RNA sequencing (RNA-Seq). Our study established a 5-day-old CVA6 infected ICR mouse model, and provided basic data for exploring the mechanism of host-cell interaction affecting viral neurogenic pathogenesis.” (page 2, lines 57-61).

- In figure 1 (line 87), it not clear which of the graphs represent which route of inoculation (intracranial, intramuscular, intraperitoneal, etc). It is implied in the figure legend, but it needs to be clearly indicated in the figure.

------ Response to reviewer 1 comment 3: Thank you for your comments, and it has been clarified in figure 1 and legend. Five-day-old ICR mice were infected with strain F219 at 103, 104, 105, and 106 TCID50 by intracranial injection (A, D and G), intraperitoneal injection (B, E and H) , and intramuscular injection (C, F and I) , respectively (n=7 per group). (Page 3, lines 91-93).

- It is not clear what "partial infection", referred to in line 75, means.

------ Response to reviewer 1 comment 4: Thank you for your comments, and it has been clarified in the manuscript. At the 104 TCID50 infection dose level, some infected mice died after i.m. and i.c. routes, and the mortality of the former was higher than that of the latter. (Page 2, lines 74-76).

- The use of "WT" to denote mock-infected mice in figures 2A and 2B is confusing, since "WT" is often used to mean "wild type".

------ Response to reviewer 1 comment 5: Thank you for your comments, and it has been clarified in the manuscript. "WT" was changed to “mock” in figures 2A and 2B.

- The legend for figure 2 says the scale bar is 100um, but that is only true for figure 2D. The scale bar for figure 2C appears to be 500um.

------ Response to reviewer 1 comment 6: Thank you for your comments, and it has been clarified in the manuscript. We confirmed the scale bar in figure 2, the magnification of Figure 2C, 2D, 2F, and 2G is Í200, and the magnification of Figure 2E and 2H is Í400, so the scale bar for Figure 2C, 2D, 2F, and 2G is100μm, and the scale bar for Figure 2E and 2H is 50μm. (Page 4,lines 112-113).

- References need to be provided for statements such as those in line 113-115, 117-118, 118-119, etc

------ Response to reviewer 1 comment 7: Thank you for your comments, and we added one reference to the content of line 113-115 (Emerg Microbes Infect. 2022 Dec; 11(1):2248-2263. A mouse-adapted CVA6 strain exhibits neurotropism and triggers systemic manifestations in a novel murine model); we added two references to the content of line 117-118 (Front Cell Infect Microbiol. 2016 Dec 21; 6: 192. The Preferential Infection of Astrocytes by Enterovirus 71 Plays a Key Role in the Viral Neurogenic Pathogenesis; and PLoS Pathog. 2019 Nov 15; 15(11):e1008142. EV71 infection induces neurodegeneration via activating TLR7 signaling and IL-6 production); and we added one reference to the content of line 118-119 (mSphere. 2021 Mar 10; 6(2):e01048-20. Pathogenesis Study of Enterovirus 71 Using a Novel Human SCARB2 Knock-In Mouse Model).

- It is not clear what the "20x" refers to in figure 3 (line 126). Is that the objective lens used? If it is, I am not sure it needs to be indicated here.

------ Response to reviewer 1 comment 8: Thank you for your comments, and it has been clarified in the manuscript. 20Xrefers the scale bar, and we have removed in the figure 3.

- It is not clear whether the heat map in figure 5H represents data from the mice tissues or the U251 cells. The legend suggests that it is both, but I don't see the point of merging them, if that is the case. In any case, the data from the brain tissue needs to be highlighted, given that the main purpose of this manuscript is to describe the mouse model being established.

------ Response to reviewer 1 comment 9: Thank you for your comments, and it has been clarified in the manuscript. The heat map in figure 5H represents the differentially expressed genes in mice brain tissue and we have clarified in the figure legend. (Page 7, line 193).

- The statement "CVA6 could cause pathogenesis at both levels to the detailed dysregulated genes..." in line 190-191 is not clear

------ Response to reviewer 1 comment 10: Thank you for your comments, and it has been clarified in the manuscript. CVA6 could cause pathogenesis in the ICR mouse brain tissues and the human glioma (U251) cells to the detailed dysregulated genes..." (Page 7, lines 195-197).

- In the Discussion section, the authors wrote "Several previous studies have reported CVA6 murine models of one- to ten-day-old BALB/c or ICR mouse strains" (line 210-211). Given that at least one previous study has shown norological and immonological phenotypes in 10-day old ICR mice (PMID: 36036059), I suggest the authors follow their statement with an explanation of how their model is superior to the previously published ones, or perhaps how it is different, what advantages it has, etc.

------ Response to reviewer 1 comment 11: Thank you for your comments,and it has been clarified in themanuscript. In this study, we established a 5-day-old CVA6 infected ICR mouse model and described the neurogenic pathogenesis. We conclude that neurotropic CVA6 preferentially infects and replicates in astrocytes of the neonatal mouse model, and also replicates well in human glioma (U251) cells. Finally, CVA6 infection increases the defense response to viral genes and reduced the number of neurological system genes in the brain and U251 cells at the transcriptome level. (Page 8, lines 212-217).

- In line 211-215, the authors state "CVA6 infection leads to neurological and systemic complications such as HFMD, aseptic meningitis, and encephalitis (9-11). In this study, the brains of infected mice showed cerebral edema, neuronal cell swelling, and high viral titers. The virus, including EV-A71, infects the human brain and has been shown to cause marked astrogliosis, microgliosis, and immune cell accumulation". This makes it sound like EV-A71 is a subset of CVA6, which is not the case.

------ Response to reviewer 1 comment 12: Thank you for your comments, and it has been clarified in the manuscript. CVA6 infection can cause neurological and systemic complications such as HFMD, aseptic meningitis, and encephalitis (9-11). In this study, the brain edema, neuronal cell swelling, and high viral titers in the infected mice will also lead to significant astrocyte proliferation, microgliosis, and immune cell aggregation. (Page 8, lines 218-221).

- In line 244-246, the authors state "...is enriched in virally infected mitochondria to promote the polyubiquitination and aggregation of mitochondrial antiviral signaling protein (MAVS)". I imagine the authors mean 'enriched in the mitochondria of virus-infected cells', since viruses don't typically infect mitochondria.

------ Response to reviewer 1 comment 13: Thank you for your comments, and revised as suggested. “is enriched in the mitochondria of virus-infected cells to promote the polyubiquitination and aggregation of mitochondrial antiviral signaling protein (MAVS)”. (Page 8, lines 249-251).

- RNA-seq data needs to be made publically available, such as by depositing it in a public repository.

------ Response to reviewer 1 comment 14: Thank you for your comments. All sequencing data in this study are available in the Sequence Read Archive (SRA) database under the accession numbers PRJNA929099 and PRJNA929115. (Page 11, lines 363-365).

Reviewer 2 Report

The authors present a mouse model of CV-A6 infection. Additionally, they assess pathogenesis using transcriptome analysis.

The study is correctly designed, the methodology is well founded, presenting descriptive results that lay a foundation for further studies. I consider that the manuscript should be accepted with some minimal comments that are made below:

- line 55. Include reference.

- line 116. Please clarify the meaning of GFAP in the paper.

- Lines 119. Include reference.

- Figure 6. I think this figure is not necessary.

- line 260. It would be interesting to clarify if the patient had neurological symptoms.

Author Response

Reviewer 2

The authors present a mouse model of CV-A6 infection. Additionally, they assess pathogenesis using transcriptome analysis.

The study is correctly designed, the methodology is well founded, presenting descriptive results that lay a foundation for further studies. I consider that the manuscript should be accepted with some minimal comments that are made below:

- line 55. Include reference.

------ Response to reviewer 2 comment 1: Thank you for your comments, and we added one reference to the content of line 55 (Expert Rev Anti Infect Ther. 2015; 13(9):1061-71. Coxsackievirus A6: a new emerging pathogen causing hand, foot and mouth disease outbreaks worldwide).

- line 116. Please clarify the meaning of GFAP in the paper.

------ Response to reviewer 2 comment 2: Thank you for your comments, and it has been clarified in the manuscript. GFAP (glial fibrillary acidic protein) is a class-III intermediate filament, a cell-specific marker, which differentiates astrocytes from other glial cells during the development of the central nervous system (Page 4, line 124-126).

- Lines 119. Include reference.

------ Response to reviewer 2 comment 3: Thank you for your comments, and we added one reference to the content of line 119 (mSphere. 2021 Mar 10; 6(2):e01048-20. Pathogenesis Study of Enterovirus 71 Using a Novel Human SCARB2 Knock-In Mouse Model).

- Figure 6. I think this figure is not necessary.

------ Response to reviewer 2 comment 4: Thank you for your comments, and we deleted the Figure 6 in the current manuscript.

- line 260. It would be interesting to clarify if the patient had neurological symptoms.

------ Response to reviewer 2 comment 5: Thank you for your comments, and it has been clarified in the manuscript. In addition to the typical clinical manifestations of HFMD, such as fever, rash on hands and feet, this patient also presented some nervous system symptoms, such as headache, poor spirit, lethargy, etc. (Page 9, lines 266-268).

Round 2

Reviewer 1 Report

Authors have improved the quality of the manuscript significantly.

Reviewer 2 Report

The authors have satisfactorily responded to the comments and suggestions made in the first version of this manuscript.